# On Higher Order Structures in Thermodynamics

**DOI:** 10.3390/e22101147

**Published:** 2020-10-12

**Authors:** Valentin Lychagin, Mikhail Roop

**Affiliations:** 1V.A. Trapeznikov Institute of Control Sciences of Russian Academy of Sciences, 65 Profsoyuznaya Str., 117997 Moscow, Russia; valentin.lychagin@uit.no; 2Faculty of Physics, Lomonosov Moscow State University, Leninskie Gory, 119991 Moscow, Russia

**Keywords:** thermodynamic states, Legendrian and Lagrangian manifolds, variance, skewness

## Abstract

We present the development of the approach to thermodynamics based on measurement. First of all, we recall that considering classical thermodynamics as a theory of measurement of extensive variables one gets the description of thermodynamic states as Legendrian or Lagrangian manifolds representing the average of measurable quantities and extremal measures. Secondly, the variance of random vectors induces the Riemannian structures on the corresponding manifolds. Computing higher order central moments, one drives to the corresponding higher order structures, namely the cubic and the fourth order forms. The cubic form is responsible for the skewness of the extremal distribution. The condition for it to be zero gives us so-called symmetric processes. The positivity of the fourth order structure gives us an additional requirement to thermodynamic state.

## 1. Introduction

The geometrical interpretation of thermodynamic systems in equilibrium goes back already to the 19th century [1] and is reflected recently in Reference [2]. In modern terms, it is clear that thermodynamic states are Legendrian submanifolds, i.e., maximal integral manifolds of the structure contact form (for more details, see Reference [3,4,5]) in the contact space where the mentioned structure form is the first law of thermodynamics. Additional structures, such as Riemannian structures on these Legendrian manifolds, were studied in, for example, Reference [6]. Considering thermodynamics in the context of measurement of random vectors [7], one gets both structures as coming from the minimal information gain or Kullback-Leibler divergence principle [8]. Namely, Legendrian manifolds represent averages of measurable quantities (or extremal probability distributions) that are extensive thermodynamic variables, while the Riemannian structure is their variance, i.e., contact and Riemannian structures arise from the first two central moments of random vectors. It is worthy of note that both these structures were widely studied, but, at the same time, the higher order structures, corresponding to central moments of higher order, have not been addressed before.

In this paper, we develop the geometrical approach to thermodynamic states and extend it by considering the third and the fourth order moments and corresponding symmetric forms of the third and the fourth order on Legendrian manifolds of two types of gases, ideal and van der Waals. The third order symmetric form represents the skewness of the extremal probability distribution, as well as thermodynamic processes along which the skewness is equal to zero, we, therefore, call *symmetric*. We elaborate such processes for ideal gases explicitly, and, in the case of real gases represented by the van der Waals model, we show that there are domains on their Legendrian manifold where there are either three types of such processes, or one. It is natural to require that the fourth order symmetric form must be positive. It is known that the positivity of the variance leads to the notion of thermodynamic phases [7], and the positivity of the fourth order form gives us an additional separation of applicable phases.

## 2. Geometry, Measurement, Thermodynamics

In this section, we briefly recall how contact geometry naturally appears in the context of measurement, as well as how symmetric *k*-forms represent *k*th central moments. For details, we refer to Reference [4,7,9].

Let (Ω,A,q) be a probability space, where Ω is a sample space, A is a σ-algebra on Ω, and *q* is a probability measure. Then, a random vector X:(Ω,A,q)→W, where *W* is a vector space, dimW=n is a measurement of $x_{0}\in W$ if $E_{q}X=x_{0}$. To measure another vector x∈W, one has to choose another measure dp=ρdq, where ρ is a probability density, such that
(1)$$E_{p}X=\int_{\Omega }X\rho dq=x,\;\int_{\Omega }\rho dq=1.$$
To find ρ, we also use the *principle of minimal information gain*:$$I\left(\rho \right)=\int_{\Omega }\rho \ln\rho dq\to \min.$$
This gives us the extremal probability distribution (see Reference [7]):$$\rho =\frac{e^{\langle \lambda ,X\rangle }}{Z(\lambda )},$$
where $\lambda \in W^{*}$, and $Z\left(\lambda \right)=\int_{\Omega }e^{\langle \lambda ,X\rangle }dq$.

Introducing the Hamiltonian H(λ)=−lnZ(λ) and using the first relation in (Equation 1), we obtain that the measurement belongs to a manifold
$$L=\left\{x=-\frac{\partial H}{\partial \lambda }\right\}\subset W\times W^{*},$$
which is Lagrangian with respect to the symplectic form
$$\omega =\sum_{i=1}^{n}d\lambda_{i}\wedge dx_{i},$$
i.e., ${\omega |}_{L}=0$. Considering the relation $x_{i}=-H_{\lambda_{i}}$, i=1,…,n as an equation for $\lambda_{i}$, one gets (locally) $\lambda_{i}=\lambda_{i}\left(x\right)$. The information gain I(x) is a function on *L* which is related with Hamiltonian H(λ) as
(2)I(x)=H(λ(x))+〈λ(x),x〉.
Therefore, we have $I_{x_{i}}=\lambda_{i}$, and, if *u* is a coordinate on R, then the submanifold
$$\hat{L}=\left\{u=I\left(x\right),\;\lambda_{i}=I_{x_{i}}\right\}\subset W\times W^{*}\times R$$
is Legendrian with respect to the contact form $\theta =du-\sum_{i=1}^{n}\lambda_{i}dx_{i}$.

The *k*th moment of random vector *X* is defined by the well-known relation
(3)$$m_{k}\left(X\right)=\int_{\Omega }X^{{⊗}^{k}}\rho dq.$$

In coordinates $\left(\lambda_{1},\ldots ,\lambda_{n}\right)$, it can be written as
(4)$$m_{k}\left(X\right)=\frac{Z_{\lambda_{i_{1}}\ldots \lambda_{i_{k}}}}{Z}d\lambda_{i_{1}}⊗\cdots ⊗d\lambda_{i_{k}}.$$
Here, and further, we use the Einstein’s summation convention. Formula (Equation 4) can be found inductively using the definition of the partition function Z(λ).

The *k*th central moment $\sigma_{k}$ is a symmetric *k*-from on *W*:(5)$$S^{k}\left(W\right)∋\sigma_{k}=\int_{\Omega }{\left(X-m_{1}\left(X\right)\right)}^{{⊗}^{k}}\rho dq.$$
Applying the standard binomial theorem, one gets the following relation between moments $m_{k}$ and central moments $\sigma_{k}$ (also see Reference [9]):(6)$$\sigma_{k}=\sum_{i=0}^{k}{\left(-1\right)}^{k-i}\left(\begin{array}{c} k\\ i \end{array}\right)m_{i}\cdot m_{1}^{{⊗}^{\left(k-i\right)}},$$
where · stands for the symmetric product.

Let us give a coordinate description of central moments of orders 2,3,4.k=2.**Theorem** **1.***On the Legendrian manifold $\hat{L}$ the second central moment has the following form in coordinates $\left(\lambda_{1},\ldots ,\lambda_{n}\right)$*(7)$$\sigma_{2}=-\frac{\partial^{2}H}{\partial \lambda_{i_{1}}\partial \lambda_{i_{2}}}d\lambda_{i_{1}}⊗d\lambda_{i_{2}},$$*and in coordinates $\left(x_{1},\ldots ,x_{n}\right)$*(8)$$\sigma_{2}=\frac{\partial^{2}I}{\partial x_{i_{1}}\partial x_{i_{2}}}dx_{i_{1}}⊗dx_{i_{2}},$$k=3.**Theorem** **2.***On the Legendrian manifold $\hat{L}$ the third central moment has the following form in coordinates $\left(\lambda_{1},\ldots ,\lambda_{n}\right)$*(9)$$\sigma_{3}=-\frac{\partial^{3}H}{\partial \lambda_{i_{1}}\cdots \partial \lambda_{i_{3}}}d\lambda_{i_{1}}⊗\cdots ⊗d\lambda_{i_{3}},$$*and in coordinates $\left(x_{1},\ldots ,x_{n}\right)$*(10)$$\sigma_{3}=-\frac{\partial^{3}I}{\partial x_{i_{1}}\cdots \partial x_{i_{3}}}dx_{i_{1}}⊗\cdots ⊗dx_{i_{3}},$$k=4.**Theorem** **3.***On the Legendrian manifold $\hat{L}$ the fourth central moment has the following form in coordinates $\left(\lambda_{1},\ldots ,\lambda_{n}\right)$*(11)$$\sigma_{4}=-\frac{\partial^{4}H}{\partial \lambda_{i_{1}}\cdots \partial \lambda_{i_{4}}}d\lambda_{i_{1}}⊗\cdots ⊗d\lambda_{i_{4}}+3\sigma_{2}\cdot \sigma_{2}.$$

Formulae (Equation 7), (Equation 9), (Equation 11) follow directly from substitution of (Equation 4)–(Equation 6) and using Z=exp(−H), while (Equation 8), (Equation 10) result from formula (Equation 2) relating the Hamiltonian H(λ) and information gain I(x). Formula for $\sigma_{4}$ in terms of $\left(x_{1},\ldots ,x_{n}\right)$ can be found, as well, but we do not provide it here because of its bulkiness.

Recall that, in thermodynamics of gases, x=(e,v)∈W, $\lambda =\left(-T^{-1},-pT^{-1}\right)\in W^{*}$, I(x)=−S(e,v), where *e* and *v* are specific inner energy and specific volume, respectively, *T* is temperature, *p* is pressure, and the contact structure is
$$\theta =-ds+T^{-1}de+pT^{-1}dv,$$
and the Legendrian manifold is given by
$$\hat{L}=\left\{s=S\left(e,v\right),\;p=\frac{S_{v}}{S_{e}},\;T=\frac{1}{S_{e}}\right\}\subset W\times W^{*}\times R.$$

It is natural to require that all the even order symmetric forms must be positive on Legendrian manifolds. In case of k=2, this condition leads to the notion of phases [4,7]. Further, we will elaborate symmetric forms of orders k=3 and k=4 in more detail for ideal and van der Waals models of gases.

It is worth mentioning that central moments $\sigma_{k}$ are preserved by the affine group Aff(W) action. In Reference [9,10], the central moments are used to construct scalar differential invariants of Aff(W).

## 3. Third Central Moment $\sigma_{3}$

The third central moment $\sigma_{3}$ is nothing but skewness of the probability distribution ρ. From the thermodynamic perspective, the symmetric 3-form $\sigma_{3}$ gives us a special type of thermodynamic processes, along which this form vanishes. From geometrical viewpoint, thermodynamic processes are considered to be contact vector fields preserving the Legendrian manifold $\hat{L}$. Assuming that these vector fields have no singular points, one may look for such processes in the form
$$X=\frac{\partial }{\partial e}+q\frac{\partial }{\partial v}.$$
Taking into account that I(x)=−S(e,v) and substituting this relation to formula (Equation 10), the condition $\sigma_{3}\left(X,X,X\right)=0$ forces the following equation on the coefficient *q*:(12)$$S_{vvv}q^{3}+3S_{evv}q^{2}+3S_{eev}q+S_{eee}=0.$$

### 3.1. Ideal Gas

For ideal gases, the entropy function is given by
$$S\left(e,v\right)=\ln\left(e^{n/2}v\right),$$
where *n* is the degree of freedom. Equation (Equation 12) in case of ideal gases takes the form
$$\frac{2q^{3}}{v^{3}}+\frac{n}{e^{3}}=0.$$
This equation has a unique real solution; therefore, in case of ideal gases, we have one type of symmetric processes shown in Figure 1. The vector field *X* takes the form
$$X=\frac{\partial }{\partial e}+{\left(\frac{n}{2}\right)}^{1/3}\frac{v}{e}\frac{\partial }{\partial v}.$$

### 3.2. van der Waals Gas

In case of van der Waals gas, we use the reduced thermodynamic variables, in terms of which the entropy function is
$$S\left(e,v\right)=\ln\left({\left(e+\frac{3}{v}\right)}^{4n/3}{\left(3v-1\right)}^{8/3}\right).$$
Here, Equation (Equation 12) is
(13)$$\begin{array}{cc} \; & (54e^{3}v^{6}-243e^{2}nv^{5}+243e^{2}nv^{4}+486e^{2}v^{5}-81e^{2}nv^{3}-\\ & -729env^{4}+9e^{2}nv^{2}+729env^{3}+1458ev^{4}-243env^{2}-729nv^{3}+\\ & +27env+729nv^{2}+1458v^{3}-243nv+27n)q^{3}+\\ & +\left(-243env^{6}+243env^{5}-81env^{4}+9env^{3}\right)q^{2}+\left(-243nv^{7}+243nv^{6}-81nv^{5}+9nv^{4}\right)q+\\ & +27nv^{9}-27nv^{8}+9nv^{7}-nv^{6}=0. \end{array}$$
In general, Equation (Equation 13) may have either three or one real roots, depending on the discriminant of the cubic (Equation 13). Therefore, we get domains where there are either three symmetric processes, or only one. They are shown in Figure 2. We use here coordinates (T,v), where *T* is temperature, instead of (e,v).

In this figure, the pink domain corresponds to that on the Legendrian manifold where there are three real roots of (Equation 13). The blue line separates the domains where the 2-form $\sigma_{2}$ is positive (above) corresponding to applicable domain (where the conditions of thermodynamic stability hold), and negative (under). This means that, in a pink part of an applicable domain of van der Waals gas, there is a family of 3 symmetric processes, while, in the white part above the blue line, only one of them remains. Thus, we can see that, in this case, there is only one transition from three symmetric processes to one.

In Figure 3, again, we are interested only in an applicable domain which is above the blue line. In this case, pink domains where there are three symmetric processes are separated from each other by white domain where there is only one symmetric process.

## 4. Fourth Central Moment $\sigma_{4}$

The analysis here is similar to that in the previous section. Consider the vector field
$$X=x_{1}\frac{\partial }{\partial e}+x_{2}\frac{\partial }{\partial v}.$$
The conditions $\sigma_{4}>0$ and $\sigma_{2}>0$ yield that the homogeneous polynomials $P_{1}\left(x_{1},x_{2}\right)=\sigma_{2}\left(X,X\right)$ and $P_{2}\left(x_{1},x_{2}\right)=\sigma_{4}\left(X,X,X,X\right)$ in $x_{1}$ and $x_{2}$ are positive.

### 4.1. Ideal Gas

In case of ideal gas, one gets
$$P_{1}\left(x_{1},x_{2}\right)=\frac{nx_{1}^{2}}{2e^{2}}+\frac{x_{2}^{2}}{v^{2}},\;P_{2}\left(x_{1},x_{2}\right)=\frac{3n(n+4)}{4e^{4}}x_{1}^{4}+\frac{3n}{v^{2}e^{2}}x_{1}^{2}x_{2}^{2}+\frac{9}{v^{4}}x_{2}^{4},$$
and both conditions hold on the entire Legendrian manifold.

### 4.2. van der Waals Gas

In this case, the standard analysis shows that the condition $\sigma_{2}>0$ holds not everywhere as well as $\sigma_{4}>0$ does. The first condition gives us separation of the Legendrian manifold to liquid and gas phases, while the second one is an additional requirement to applicable with respect to $\sigma_{2}$ states. This is shown in Figure 4. In the pink domain, both conditions $\sigma_{2}>0$ and $\sigma_{4}>0$ hold. The red line separates domains where $\sigma_{2}$ has opposite signs.

We can see that the positivity of $\sigma_{4}$ gives us an additional clarification of the applicability of the van der Waals model.

## 5. Discussion

This paper presents some new results in equilibrium thermodynamics that arise from the measurement approach to thermodynamics. By measuring random vectors, components of which in thermodynamics are specific energy and volume, one gets a Legendrian manifold representing the average of random vector. The variance $\sigma_{2}$ of random vectors leads to the Riemannian structure on the corresponding Legendrian manifold, and condition of positivity of $\sigma_{2}$ forces a well-known condition of thermodynamic stability of a thermodynamic system in equilibrium. The next natural step is to elaborate higher order central moments, for instance, $\sigma_{3}$, which is skewness, and $\sigma_{4}$. They lead to the third and the fourth order symmetric forms on the Legendrian manifold, similarly to that of the second order induced by the second central moment $\sigma_{2}$. It is clear that the even order forms should be positive on the thermodynamic Legendrian manifolds, which can be interpreted by the following way. Not all the points on the Legendrian manifolds correspond to real physical states, but only those of them where $\sigma_{2}$, $\sigma_{4}$, and so on are positive. Here, we have considered only $\sigma_{4}$, but even this additional (comparing with traditional ones) structure has lead us to new applicable domains on van der Waals thermodynamic Legendrian manifold, while, for an ideal gas, $\sigma_{4}$ is positive everywhere. Namely, we can see from Figure 4 that the critical point $\left(T_{\mathrm{crit}},v_{\mathrm{crit}}\right)=\left(1,1\right)$ no longer belongs to the domain where both $\sigma_{2}$ and $\sigma_{4}$ are positive. Instead of this, we get something like a new critical point, which is the maximum of the curve separating domains where $\sigma_{4}>0$ and $\sigma_{4}<0$. Note that this curve is of the similar form as that separating domains where $\sigma_{2}>0$ and $\sigma_{2}<0$. This means that we get a more accurate applicability condition for van der Waals model, and it may be a considerable contribution to the theory of phase transitions, since, as we know, it is the sign changing of $\sigma_{2}$ that forces the first order phase transitions in van der Waals model [4,5,7].

We may say that applicable domain of real gases is an intersection of areas where all even central moments are positive. We could see that, by adding the fourth central moment, we shrink an applicable domain of the van der Waals model comparing with well-known one where $\sigma_{2}>0$. The natural question is will there remain any domains where all even central moments are positive. Our hypothesis is that there will be such domains, but theoretical proof of this statement is an open question.

## Figures and Tables

**Figure 1 entropy-22-01147-f001:**
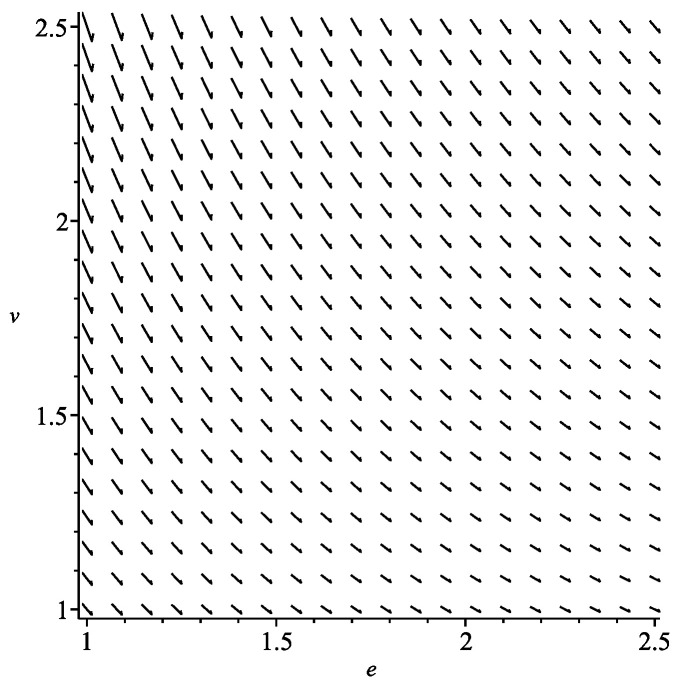
Symmetric process for ideal gas.

**Figure 2 entropy-22-01147-f002:**
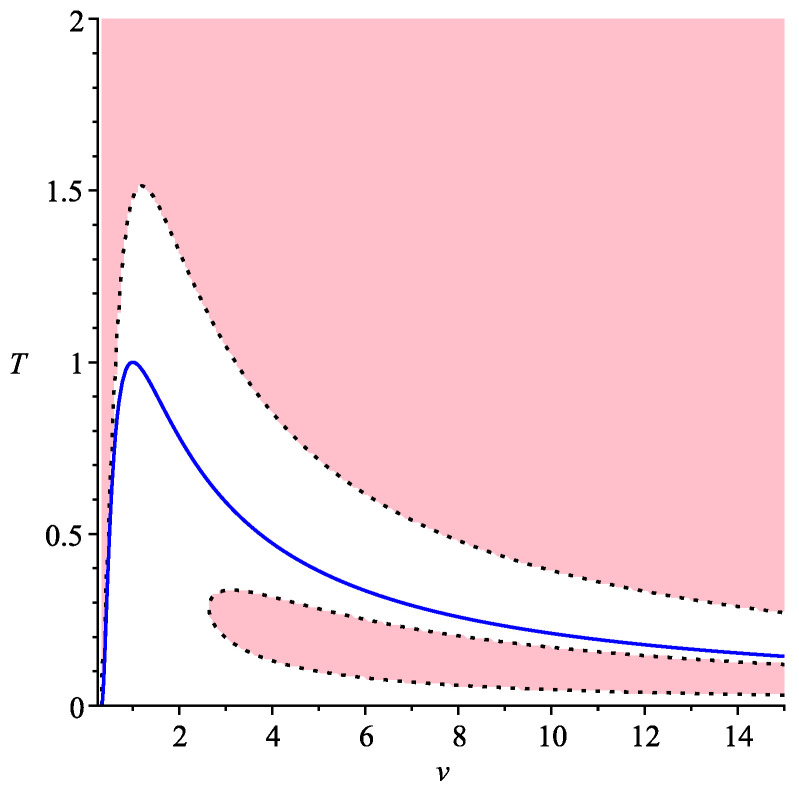
Symmetric processes for van der Waals gas, n=3.

**Figure 3 entropy-22-01147-f003:**
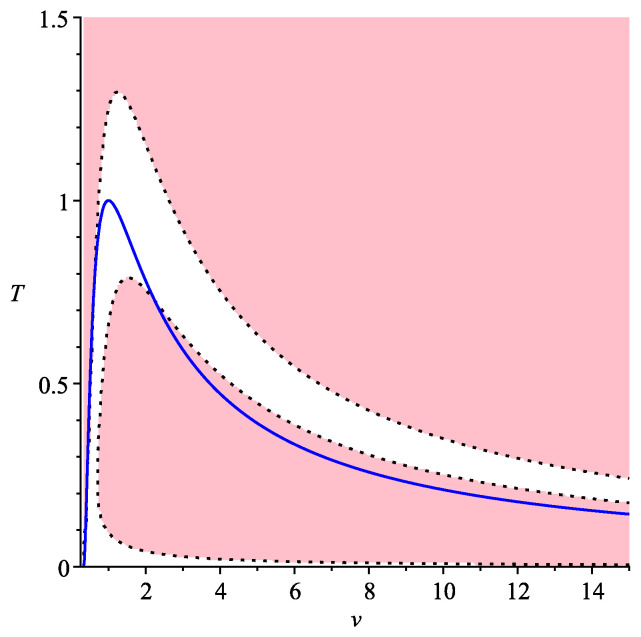
Symmetric processes for van der Waals gas, n=13.

**Figure 4 entropy-22-01147-f004:**
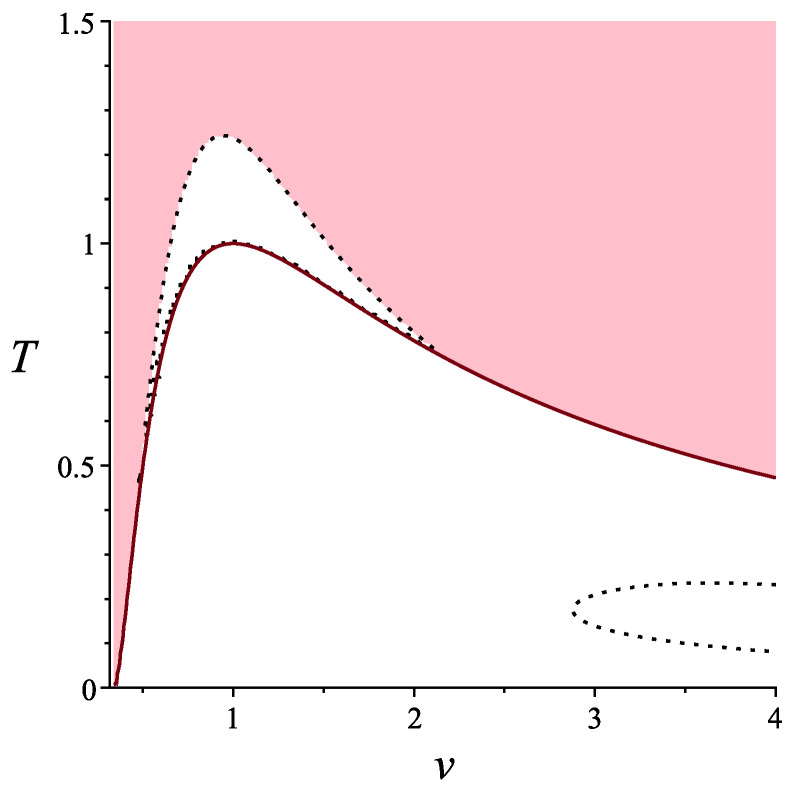
Applicable domains for van der Waals gas, n=3.

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
