# Peer review of "On Higher Order Structures in Thermodynamics"

_entropy, 2020, doi:10.3390/e22101147_

Round 1

Reviewer 1 Report

The work presented by Lychagin and Roop is an attempt to describe some thermodynamic properties using a geometrical approach. In particular, they are interested in the application of measure theory for the describing of the third and fourth moments of probability distributions with thermodynamic meaning.

Geometrical approaches are, indeed, a very useful tool for the deep understanding of the mathematical structure of thermodynamics. However, I am afraid that the present work lacks of clarity, which inhibits the assessment of the manuscript's scientific contribution.

I have several problems with the structure of the manuscript. In the introduction, the concepts of both Legendrian and Lagrangian manifolds, as well as Riemannian structures are presented as standard knowledge. Despite of Entropy is a specialized journal in mathematical and theoretical thermodynamics, I'm not sure that such concepts are familiar for all the target readers. A brief discussion about the definition and importance of these mathematical concepts and their connection with thermodynamic parameters or functions would be very appreciated by the reader. I found very hard to evaluate the relevance of the second section, I understand that the methodology is presented in this section, but I can not see clearly the connection with the section three. Most of the equations in section two are barely related with the results contained in the following sections . The results in sections 3 and 4 are presented without clarity. For example, the expressions for the third moment of both ideal gas and van der Waals fluid are presented only referring to Eq. (2). However, such equation is very hard to understand and even more difficult to apply. Besides, the results are not discussed in detail, it is impossible to make a good appreciation of the presented equations without a remark about its conceptual importance or applicability. Finally, the manuscript lacks of conclusions.

In addition, I have some conceptual problems with the manuscript. For instance, accordingly to standard statistical thermodynamics, the equilibrium states always have symmetric probability functions. Furthermore, is well known that equilibrium systems obey gaussian distribution functions. What is the meaning of the multiple solutions for the Eq. (2)? there exist other functional forms for probability distributions consistent with equilibrium states?

What is the meaning of the second equation of section 3.1?, I mean, is this equation consistent with the expected behavior of an ideal gas?, Which is its mathematical or physical meaning?, Why is important such equation?. Same questions for Eq. (3).

The statement in section 3.2 (line 53) "Therefore we get domains where there are either three symmetric processes, or only one.", requires a detailed discussion about the meaning and the relevance of these symmetric processes in the context of equilibrium thermodynamics.

For the reasons exposed previously, I do not recommend the manuscript for publication.

Author Response

Please find the authors' response in the attached file.

Reviewer 2 Report

Authors develop the geometrical approach to thermodynamical states. The geometrical interpretations of thermodynamics were widely studied. Nevertheles central moments of higher order have not been addressed before.

The paper is interesting and worth publication and broad discussion.

Please check definition of k-th central moment (p.3); typos ? or some kind of notion. If possible, some example of applications will be valuable. Summary (concluding remarks for broad spectrum of readers) of obtained results is needed.

In my opinion the paper (after small revision) can be recommended for publication.

Author Response

Dear Reviewer 2,

Thank you for your attention to the paper.

Definition (5) for the k-th central moment seems correct. We use here tensor product, which differs this definition from the usual coordinate one. We added section "Discussion", where we clarify obtained results and highlight further developments.

Reviewer 3 Report

The Authors develop a geometrical approach to thermodynamics, and focus in particular on in the third and fourth order moments of the probability distribution. Even if the results sound correct, in my opinion sections 3 and 4, which contain the novel contributions of this manuscript, should be slightly extended and the results and plots should be better discussed. Apart from this, in my opinion the manuscript deserves publication on this journal.

Author Response

Dear Reviewer 3,

Thank you for your attention to the paper.

In the revised version we made the exposition of the results clearer and added section "Discussion", where we explain the significance of the results.

Round 2

Reviewer 1 Report

The new version of the manuscript presented by Lychagin and Roop is a good improvement with respect the first draft. Despite the are not much changes, the new content represents an overall improvement in the clarity of the paper. 

In contrast with my first appreciation, the new version contains enough information to consider it self-contained. I believe that a motivated reader can find the necessary background to understand the content of the manuscript. 

I feel comfortable with the final version and, in consequence, I do recommend for publication in Entropy.